# Scarring in Rough Rectangular Billiards

**DOI:** 10.3390/e25020189

**Published:** 2023-01-18

**Authors:** Felix M. Izrailev, German A. Luna-Acosta, J. A. Mendez-Bermudez

**Affiliations:** 1Instituto de Física, Benemérita Universidad Autónoma de Puebla, Apartado Postal J-48, Puebla 72570, Mexico; 2Department of Physics and Astronomy, Michigan State University, East Lansing, MI 48824-1321, USA

**Keywords:** scars, localization effects, quantum billiards, rough billiards

## Abstract

We study the mechanism of scarring of eigenstates in rectangular billiards with slightly corrugated surfaces and show that it is very different from that known in Sinai and Bunimovich billiards. We demonstrate that there are two sets of scar states. One set is related to the bouncing ball trajectories in the configuration space of the corresponding classical billiard. A second set of scar-like states emerges in the momentum space, which originated from the plane-wave states of the unperturbed flat billiard. In the case of billiards with one rough surface, the numerical data demonstrate the repulsion of eigenstates from this surface. When two horizontal rough surfaces are considered, the repulsion effect is either enhanced or canceled depending on whether the rough profiles are symmetric or antisymmetric. The effect of repulsion is quite strong and influences the structure of all eigenstates, indicating that the symmetric properties of the rough profiles are important for the problem of scattering of electromagnetic (or electron) waves through quasi-one-dimensional waveguides. Our approach is based on the reduction of the model of one particle in the billiard with corrugated surfaces to a model of two artificial particles in the billiard with flat surfaces, however, with an effective interaction between these particles. As a result, the analysis is conducted in terms of a two-particle basis, and the roughness of the billiard boundaries is absorbed by a quite complicated potential.

## 1. Introduction

Recently, there has been a sharp increase in articles discussing the occurrence of ergodicity in physical systems, both classical and quantum [1,2,3,4,5,6,7,8]. This interest is due to the practical problem of statistical description of the behavior of a closed many-particle system whose equations of motion are strictly deterministic, that is, do not contain any random parameters. For a long time, this problem was considered fundamental for the justification of statistical mechanics, and it was assumed that its solution is associated with the so-called ergodicity of the motion of the system under consideration. The classical trajectory of ergodic systems densely and uniformly covers the entire phase space of the system, as a result of which the time average value of any observable is equal to the phase space average as time tends to infinity.

As a result of numerous attempts to prove this property in general terms for real physical systems, it was realized that the proof is possible only for special systems. The most well-known of such systems are the so-called scattering billiards, in particular, Sinai and Bunimovich billiards, in which the trajectory of a particle due to multiple elastic reflections from the walls is unstable for any initial condition except for special values whose measure is equal to zero. Thus, the behavior of the system is determined by a single trajectory, which makes it possible to speak of an ergodic type of motion.

It should be noted that the instability of motion in such billiards is exponentially strong, at which the distance between two close trajectories grows, on average, exponentially fast with time. When trajectories are reflected from the walls of the billiard, the so-called mixing occurs, as a result of which the motion becomes chaotic and indistinguishable from random. Indeed, as already discussed by the Soviet physicist N. S. Krylov in his book [9], the mixing is the main mechanism for the emergence of the most characteristic property of the statistical behavior, namely, the relaxation of a system to statistical equilibrium. Conversely, examples can be given when a given system is ergodic but its movement is regular, meaning that the relaxation of the system occurs at infinite times, which does not correspond to physical reality.

For quantum systems, the situation is much more complicated, since there is no rigorous and generally accepted definition of quantum ergodicity. The only reliable situation when one can speak of quantum ergodicity is for the systems that are fully ergodic in the classical limit. Needless to say, there are just a few such systems. As an example of the “true” quantum ergodicity, one can consider completely random matrices of large size for which the theory is well developed. For such matrices, the components of the eigenfunctions are Gaussian distributed, which is a property that is a consequence of the uniform distribution of the eigenfunction vector over the surface of the *N*-dimensional sphere with N≫1. It can be seen that in this case, the eigenfunctions are not only ergodic but also maximally chaotic (random), see e.g., [10].

However, for realistic physical systems, many-body matrix elements in a physically chosen basis, as a rule, are not completely random and fill only a band of finite width (determined by a finite radius of interaction in the energy space). Therefore, one can speak of ergodic eigenfunctions occupying only some energy shell of finite size. Clearly, determining the shape and size of this shell is extremely difficult [11]. In any case, at the moment, there is no unified theory of quantum ergodicity of isolated systems which takes into account the influence of the finiteness of the energy shell.

Returning to systems with a small number of particles (small number of degrees of freedom), for example chaotic billiards for which classical ergodicity is rigorously proven, after several years of their study, it was found that many of the eigenfunctions are non-ergodic even for very high energies [12,13]. In principle, such non-ergodic states do not contradict Shnirel’man’s theorem [14], according to which their number must decrease with increasing energy. Numerical experiments with ergodic billiards have shown that in many cases, these “scarring” states in the configuration space can be associated with unstable periodic orbits, the number of which increases with energy. However, since these trajectories are isolated, when averaged over infinite time, they do not contribute to the mean values of the observables.

To date, there have been many papers that investigate the properties of many-particle scarring states in various physical systems (see, for example, [15,16,17,18,19,20,21] and references therein). The main interest is related to the mechanism for the emergence of such states immersed in a set of ergodic states. In the case of quantum systems that do not have a corresponding classical analogue, this issue becomes extremely complicated. It was found that the main characteristic of such scars in the quantum description is a small number of components of the corresponding eigenstates treated on some physically justified basis in comparison with the large number of states that can be considered as ergodic. Thus, scars can be defined as the localized states, associated with unstable periodic orbits when applicable, and embedded into the set of the ergodic states. As a result of numerous studies of quantum many-body systems, it became clear that the typical mechanism for scars is the presence of local symmetries of the system, which can correspond to the presence of local integrals of motion [20,21].

Our interest in this article is to study the mechanism for the emergence of scar states in rectangular billiards with rough horizontal surfaces. In these billiards, it is relatively easy to trace their emergence since such systems can be considered close to integrable, despite the fact that they are seem to be rigorously ergodic even for any weak boundary roughness. Such billiards of finite size were under close investigation in view of surface scattering in quasi-one-dimensional waveguides. The conventional theory of scattering in the waveguides with a weak roughness was developed long ago; however, recently, researchers found a new mechanism of surface scattering which is due to the so-called square-gradient roughness [22,23,24,25]. This mechanism was neglected in previous studies; however, it should be taken into account when the correlation length computed along the scattering profiles is very small. In particular, it was analytically shown that the scattering properties strongly depend on the correlations between upper and lower horizontal profiles. Specifically, the scattering is very different whether the two random profiles are symmetric or antisymmetric. Our study also shows a quite strong dependence of the degree of localization of the scar states on the type of correlations between the scattering profiles.

In Section 2, we describe the model under study, giving exact expressions for the matrix elements of an effective Hamiltonian in close correspondence with the model. In Section 3, we explain how we measure the degree of localization of the eigenstates in the energy basis corresponding to non-interacting artificial particles and therefore to the billiard with flat horizontal surfaces. In this representation, one easily identifies one set of scar states related to classical trajectories which are parallel to the rough boundaries of the billiard. We also show what these eigenstates look like in the configuration representation for one rough surface. In addition, we demonstrate that there is another set of scar states related to bouncing ball classical trajectories which are perpendicular to the rough boundaries of the billiard. These states consist of many “unperturbed eigenstates”, which is in contrast with those from the first set. In Section 5, we analytically show how the number of strongly localized states from the first set, which corresponds to classical plane waves with the longest perpendicular wavelength, decreases with energy. In Section 6, we study billiards with two horizontal rough profiles and show how the structure of localized eigenstates depends on whether the profiles are symmetric or antisymmetric. The main attention here is paid to the effect of the repulsion of eigenstates from the corrugated surfaces. In Section 7, we draw our conclusions.

## 2. Model of Rough Billiards

We consider billiards that are periodic in the longitudinal coordinate *x*, with Dirichlet boundary conditions on the upper $f_{1}$ and lower $f_{2}$ surfaces,
(1)$$\begin{array}{ccc} f_{1} & = & L_{y}+W_{1}\xi_{1}\left(x\right)\;,\\ f_{2} & = & W_{2}\xi_{2}\left(x\right). \end{array}$$
Here, $L_{y}$ is the average width of the billiard and $\xi_{1,2}\left(x+L_{x}\right)=\xi_{1,2}\left(x\right)$ with $\langle \xi_{1,2}\left(x\right)\rangle =0$. The angular brackets stand for the average over one period $L_{x}$ or, in the case of a random profile, over different realizations of $\xi_{1,2}\left(x\right)$.

This model has been thoroughly studied in refs. [26,27,28,29,30,31,32,33,34,35] for the specific case $\xi_{1}\left(x\right)=\cos\left(2\pi x/L_{x}\right)$ and $\xi_{2}\left(x\right)=0$, the so-called “cosine” or “ripple” billiard [36]. The main interest was in the properties of the energy spectrum [26,28] and in the quantum–classical correspondence for the shape of eigenstates (SE) and local density of states (LDOS) [29,30,31,32]. It was shown that for highly excited states, the global properties of the SES and LDOS in the quantum regime are similar to those described by the equations of motion for a classical particle moving inside the billiard. On the other hand, quite strong quantum effects have been revealed for individual eigenstates in a deep semiclassical region [31,32].

In this paper, we address the case of billiards with rough surfaces,
(2)$$\xi_{1,2}\left(x\right)=\sum_{k=1}^{N_{T}}A_{k}\cos\left(\frac{2\pi kx}{L_{x}}\right),$$
focusing on the structure of eigenstates, see also [22,23,24,25,37,38,39]. The surfaces are modeled by a large sum of harmonics $A_{k}$ drawn from a flat random distribution defined in the interval [−A,A] (with A such that $|\xi_{1,2}\left(x\right)|\leq 1$). With the increase of $N_{T}$, the degree of complexity of $\xi_{1,2}\left(x\right)$ also increases, and for $N_{T}\gg 1$, the surfaces can be treated as random: see an example in Figure 1. In what follows, we focus on rough billiards with $N_{T}=100$. For simplicity, we choose $\xi_{1}\left(x\right)=\xi_{2}\left(x\right)$. In the first part of our paper, we review the case of billiards with one flat boundary; that is $W_{2}=0$, see, e.g., Figure 1. Below, in Section 6, we consider the billiards with two antisymmetric, $W_{1}=W_{2}$, and symmetric surfaces, $W_{1}=-W_{2}$.

Originally, the model is described by the Hamiltonian
(3)$$\hat{H}=\frac{1}{2m}\left({\hat{P}}_{x}^{2}+{\hat{P}}_{y}^{2}\right)=-\frac{\hbar^{2}}{2m}\left(\partial_{x}^{2}+\partial_{y}^{2}\right),$$
for a free particle inside the billiard with rough boundaries (such that the corresponding wave function Ψ(x,y) obeys Dirichlet boundary conditions, Ψ(x,y)=0 at $y=f_{1}\left(x\right)$ and $y=f_{2}\left(x\right)$). However, in order to solve it numerically, it is useful to make a canonical transformation to new variables in which the new Hamiltonian incorporates surface-scattering effects into an effective interaction potential between artificial particles identified with the two new degrees of freedom. This can be achieved by the transformation to new canonical coordinates,
(4)$$\begin{array}{ccc} u & = & x\;,\\ v & = & \frac{f_{2}\left(x\right)-y}{f_{2}\left(x\right)-f_{1}\left(x\right)}. \end{array}$$
As a result, the boundary conditions for the new wave function are trivial: Ψ(u,v)=0 at v=0 and v=1.

The Schrödinger equation in new coordinates can be obtained from the covariant expression for a particle moving (in the absence of potentials) in a Riemannian curved space [40],
(5)$$-\frac{\hbar^{2}}{2m}\Delta_{cov}\Psi \left(u,v\right)=\frac{\hbar^{2}}{2m}g^{-1/2}\partial_{\alpha }g^{\alpha \beta }g^{1/2}\partial_{\beta }\Psi \left(u,v\right)\;,$$
where the quantum Hamiltonian in covariant form [41] is
(6)$$\hat{H}=\frac{1}{2m}g^{-1/4}{\hat{P}}_{\alpha }g^{\alpha \beta }g^{1/2}{\hat{P}}_{\beta }g^{-1/4}\;,$$
and the covariant momenta are
(7)$${\hat{P}}_{\alpha }=-i\hbar \left[\partial_{\alpha }+\frac{1}{4}\partial_{\alpha }ln\left(g\right)\right]=-i\hbar g^{-1/4}\partial_{\alpha }g^{1/4}.$$
Here, α,β=u,v, the operator $\Delta_{cov}$ is the Laplace–Beltrami operator, *g* is the metric, and $g^{\alpha \beta }$ is the metric tensor. The wave functions Ψ(u,v) are normalized as
(8)$$\int_{0}^{L_{x}}du\int_{0}^{L_{y}}dv\sqrt{g}\Psi^{†}\Psi \;=1.$$

Even though (Equation 5) is still the kinetic energy, the resulting differential equation takes a much more complicated form than the ordinal Laplacian. This is the price to pay when transferring the effect of the boundaries onto the operator. Then, by substituting in (Equation 6) the explicit expressions for the metric tensor
$$g^{\alpha \beta }=\left(\begin{array}{cc} 1 & -\frac{f_{1}^{\prime }v+f_{2}^{\prime }\left(1-v\right)}{f_{1}-f_{2}}\\ -\frac{f_{1}^{\prime }v+f_{2}^{\prime }\left(1-v\right)}{f_{1}-f_{2}} & \frac{1+{\left[f_{1}^{\prime }v+f_{2}^{\prime }\left(1-v\right)\right]}^{2}}{{\left(f_{1}-f_{2}\right)}^{2}} \end{array}\right)\;,$$
where $f_{1,2}^{\prime }\equiv \partial f_{1,2}\left(u\right)/\partial u$, and the metric
(9)$$g=Det\left(g_{\alpha \beta }\right)={\left(f_{1}-f_{2}\right)}^{2}\;,$$
we obtain
(10)$$\begin{array}{ccc} \hat{H} & = & \frac{1}{2m}g^{-1/4}\left\{{\hat{P}}_{u}\left(f_{1}-f_{2}\right){\hat{P}}_{u}+{\hat{P}}_{v}\frac{1+{\left[f_{1}^{\prime }v+f_{2}^{\prime }\left(1-v\right)\right]}^{2}}{{\left(f_{1}-f_{2}\right)}^{2}}{\hat{P}}_{v}\right.\\ & & -\left.\left[{\hat{P}}_{u}\left(f_{1}^{\prime }v+f_{2}^{\prime }\left(1-v\right)\right){\hat{P}}_{v}+{\hat{P}}_{v}\left(f_{1}^{\prime }v+f_{2}^{\prime }\left(1-v\right)\right){\hat{P}}_{u}\right]\right\}g^{-1/4}. \end{array}$$
Note that the rough boundaries $f_{1,2}$ and their derivatives $f_{1,2}^{\prime }$ are fully incorporated into the Hamiltonian operator.

We remark that this representation allows us to treat the original model of one free particle in the rough billiard as a model of two interacting “particles” identified with the two degrees of freedom *u* and *v*, where the Hamiltonian (Equation 11) is separated as
(11)$$\hat{H}={\hat{H}}^{0}+\hat{V}\left(u,v,{\hat{P}}_{u},{\hat{P}}_{v}\right)\;,$$
with
(12)$${\hat{H}}^{0}=\frac{1}{2m}\left({\hat{P}}_{u}^{2}+{\hat{P}}_{v}^{2}\right).$$
Here, ${\hat{P}}_{u}=-i\hbar \left[\partial_{u}+\left(1/4\right)\partial_{u}\ln\left(g\right)\right]$, ${\hat{P}}_{v}=-i\hbar \partial_{v}$, and $\hat{V}$ stands for an effective interaction potential between the “particles” *u* and *v*. In the following, we treat ${\hat{H}}^{0}$ as the Hamiltonian of two *non-interacting particles*. The eigenstates of ${\hat{H}}^{0}$ define the unperturbed basis in which the eigenstates of the total Hamiltonian $\hat{H}\left(u,v,{\hat{P}}_{u},{\hat{P}}_{v}\right)$ are expanded. Such a representation turns out to be convenient for the study of the chaotic properties of the model, since one can use the tools and concepts developed in the theory of interacting particles (see, for example, [42,43]).

Now, we expand the αth eigenstate of energy $E^{\alpha }$ as
(13)$$\Psi^{\alpha }\left(u,v\right)=\sum_{m}\sum_{n}C_{mn}^{\alpha }\phi_{mn}\left(u,v\right)\;,$$
where $\phi_{mn}\left(u,v\right)=\langle u,v|m,n\rangle$ are the eigenstates of the unperturbed Hamiltonian ${\hat{H}}^{0}\left(u,v\right)$, and the indexes *n* and *m* are the quantum numbers corresponding to the *u* and *v* coordinates, respectively. The function $\phi_{mn}\left(u,v\right)$ has to satisfy the boundary conditions of the problem in order to form a Galerkin basis [44]. The eigenstates of ${\hat{H}}^{0}$ are
(14)$$\phi_{mn}\left(u,v\right)=\frac{1}{\pi^{1/2}g^{1/4}}\sin\left(\frac{m\pi v}{L_{y}}\right)\exp\left[i\left(k+\frac{2\pi n}{L_{x}}\right)u\right]\;,$$
with eigenvalues
(15)$$E_{mn}^{0}\left(k\right)=\frac{\hbar^{2}}{2m}\left[{\left(k+\frac{\pi n}{L_{x}}\right)}^{2}+{\left(\frac{\pi m}{L_{y}}\right)}^{2}\right].$$
Here, formally, −∞<n<∞ and 1≤m<∞, but in practice, for the numerical computations, the unperturbed basis is truncated such that $-N_{max}<n<N_{max}$ and $1\leq m<M_{max}$. The factor $\pi^{-1/2}g^{-1/4}$ in (Equation 15) arises from the orthonormality condition (Equation 8).

Then, we can solve the eigenvalue problem by diagonalizing the Hamiltonian (Equation 11) on the basis $\phi_{mn}\left(u,v\right)$:(16)$$\sum_{m^{\prime }}\sum_{n^{\prime }}H_{mn,m^{\prime }n^{\prime }}\phi_{m^{\prime }n^{\prime }}\left(u,v\right)=E\phi_{mn}\left(u,v\right)\;,$$
where the matrix elements are
$$\begin{array}{ccc} & & H_{mn,m^{\prime }n^{\prime }}\equiv \langle mn|\hat{H}|m^{\prime }n^{\prime }\rangle \\ & & =-\frac{\hbar^{2}}{2m}\int_{0}^{L_{x}}du\int_{0}^{L_{y}}dv\;g^{1/2}\left[\phi_{mn}^{†}\right]\Delta_{cov}\left[\phi_{m^{\prime }n^{\prime }}\right]=-\frac{\hbar^{2}}{2m}\int_{0}^{L_{x}}du\int_{0}^{L_{y}}dv\;\partial_{\alpha }\left[\phi_{mn}^{†}\right]g^{1/2}g^{\alpha \beta }\partial_{\beta }\left[\phi_{m^{\prime }n^{\prime }}\right]. \end{array}$$

Since the Hamiltonian (Equation 11) is periodic in *u*, its eigenstates are Bloch states. This allows us to write the solution of the Schrödinger equation in the form $\Psi \left(u,v\right)=\exp\left(iku\right)\;\Psi_{k}\left(u,v\right)$, with $\Psi_{k}\left(u+L_{x},v\right)=\Psi_{k}\left(u,v\right)$. Here, the Bloch wave vector k(E) is in the first Brillouin zone, ($-\pi /L_{x}\leq k\leq \pi /L_{x})$. The statistical properties of eigenstates do not depend on a specific value of the Bloch index *k* inside the band, except at k=0 and $k=\pm \pi /L_{x}$, so we avoid these values of *k* in our numerical calculations.

Finally after some algebra, we obtain
(17)$$\begin{array}{ccc} & & H_{mn,m^{\prime }n^{\prime }}=\;\;\frac{\hbar^{2}}{2m}\left\{{\left(k+\frac{\pi n}{L_{x}}\right)}^{2}\delta_{nn^{\prime }}\delta_{mm^{\prime }}+\frac{m^{2}\pi^{2}}{L_{x}}\left[J_{1}+J_{2}+\left(\frac{1}{3}+\frac{1}{4m^{2}\pi^{2}}\right)J_{3}\right]\delta_{mm^{\prime }}\right.\\ & & +\left.i\;\frac{4mm^{\prime }}{\left(m^{2}-m^{\prime 2}\right)L_{x}}\left(k+\frac{\pi (n+n^{\prime })}{L_{x}}\right)\left[J_{4}-{\left(-1\right)}^{m+m^{\prime }}J_{5}\right]\;+\;\frac{4mm^{\prime }\left(m^{2}+m^{\prime 2}\right)}{{\left(m^{2}-m^{\prime 2}\right)}^{2}L_{x}}\left[{\left(-1\right)}^{m+m^{\prime }}J_{6}-J_{7}\right]\right\} \end{array}$$
where
$$\begin{array}{ccc} J_{1} & = & \int_{0}^{L_{x}}dx\;\exp\left[-i\frac{2\pi }{L_{x}}\left(n-n^{\prime }\right)x\right]\frac{f_{1}^{\prime }f_{2}^{\prime }}{{\left(f_{1}-f_{2}\right)}^{2}}\;,\\ J_{2} & = & \int_{0}^{L_{x}}dx\;\exp\left[-i\frac{2\pi }{L_{x}}\left(n-n^{\prime }\right)x\right]\frac{1}{{\left(f_{1}-f_{2}\right)}^{2}}\;,\\ J_{3} & = & \int_{0}^{L_{x}}dx\;\exp\left[-i\frac{2\pi }{L_{x}}\left(n-n^{\prime }\right)x\right]\frac{{\left(f_{1}^{\prime }-f_{2}^{\prime }\right)}^{2}}{{\left(f_{1}-f_{2}\right)}^{2}}\;,\\ J_{4} & = & \int_{0}^{L_{x}}dx\;\exp\left[-i\frac{2\pi }{L_{x}}\left(n-n^{\prime }\right)x\right]\frac{f_{2}^{\prime }}{f_{1}-f_{2}}\;,\\ J_{5} & = & \int_{0}^{L_{x}}dx\;\exp\left[-i\frac{2\pi }{L_{x}}\left(n-n^{\prime }\right)x\right]\frac{f_{1}^{\prime }}{f_{1}-f_{2}}\;,\\ J_{6} & = & \int_{0}^{L_{x}}dx\;\exp\left[-i\frac{2\pi }{L_{x}}\left(n-n^{\prime }\right)x\right]\frac{\left(f_{1}^{\prime }-f_{2}^{\prime }\right)f_{1}^{\prime }}{{\left(f_{1}-f_{2}\right)}^{2}}\;,\\ J_{7} & = & \int_{0}^{L_{x}}dx\;\exp\left[-i\frac{2\pi }{L_{x}}\left(n-n^{\prime }\right)x\right]\frac{\left(f_{1}^{\prime }-f_{2}^{\prime }\right)f_{2}^{\prime }}{{\left(f_{1}-f_{2}\right)}^{2}}. \end{array}$$
Notice that depending on the billiard geometry, some of the integrals above may vanish. For example, in the case $W_{2}=0$ (i.e., for the billiard of Figure 1), we have $J_{1}=J_{4}=J_{7}=0$.

## 3. Structure of the Hamiltonian Matrix

In order to study the structure of eigenstates of the total Hamiltonian $\hat{H}\left(u,v\right)$ one needs, first, to choose a way of ordering the unperturbed basis in which we represent the Hamiltonian matrix $H_{l,l^{\prime }}\left(k\right)={\langle l|\hat{H}|l^{\prime }\rangle }_{k}$. Specifically, we have to assign an index *l*, labeling the basis state ${|l\rangle }_{k}{\equiv |m,n\rangle }_{k}$ to each pair of indecies (m,n) (note that although the energy spectra are independent of the assignment (m,n)→l, the structure of the eigenstates is not). In addition, note that the size of the Hamiltonian matrix is determined by the maximum values of *n* and *m*: $-N_{max}\leq n\leq N_{max}$ and $1\leq m\leq M_{max}$.

A natural assignment is the following one [30]. Let us fix the lowest value of *n*$\left(-N_{max}\right)$ and sweep all values of $m\;(1\leq m\leq M_{max})$. This gives $l=1,2,\ldots ,M_{max}$. Then, do the same for $n=-N_{max}+1$, which gives $M_{max}+1\leq l\leq 2M_{max}$, and so on, until finally, we have $1\leq l\leq L_{max}$, where $L_{max}=\left(2N_{max}+1\right)M_{max}$ defines the Hamiltonian matrix size, $L_{max}\times L_{max}$. This rule results in a block structure of the Hamiltonian matrix with a block size equal to $M_{max}$. Figure 2 (left) shows the lower part of a 4030×4030 matrix with $\left(N_{max},M_{max}\right)=\left(32,62\right)$. Here, we can see a number of blocks of size 62×62 corresponding to $n,n^{\prime }=-32,\ldots ,-17$. In this representation, the matrix is clearly a block matrix.

The above way of ordering the unperturbed basis corresponds to the “channel representation” (or momentum representation), since the index *m* labels a specific transverse channel for the propagation of the wave through the billiard; see Equation (Equation 15). However, for our purposes, it is essential to use the “energy representation”, according to which the unperturbed basis is ordered in increasing energy, $E_{l_{new}+1}^{0}\left(k\right)\geq E_{l_{new}}^{0}\left(k\right)$. This defines a new rule $l\to l_{new}=l_{new}\left(n,m\right)$. In Figure 2 (right), we show the lower part of the Hamiltonian matrix of Figure 2 (left) but now in the “energy representation”. Notice that in contrast to the block structure of the Hamiltonian matrix in the “channel representation”, in the “energy representation”, the Hamiltonian matrix shows a band-like structure.

In order to analyze the structure of the eigenstates of $\hat{H}\left(u,v\right)$ in detail, we diagonalize Hamiltonian matrices in the “energy representation” and construct the “state matrices” $|C_{l_{new}}^{\alpha }{|}^{2}$. In Figure 3, we show the lower part of the state matrix corresponding to the Hamiltonian matrix of Figure 2 (right). Here, $C_{l_{new}}^{\alpha }$ are the amplitudes of the eigenstates in the (energy-ordered) basis representation given by the index $l_{new}$. Namely, the index $l_{new}$ refers to unperturbed basis states that correspond to the unperturbed Hamiltonian ${\hat{H}}^{0}$. The index α refers to a specific exact eigenstate. All eigenstates are ordered in increasing energy, with α=1 the ground state. Therefore, to understand how strongly localized/extended the exact eigenstates in the unperturbed basis are, one should fix the value of α and explore the dependence of $|C_{l_{new}}^{\alpha }{|}^{2}$ on $l_{new}$.

A crucial point in our study is that the eigenstates of the Hamiltonian in the “energy representation” have a very convenient form for the analysis. The advantage of the “energy representation” over the “channel representation” (i.e., when the Hamiltonian matrix has a block structure) is clearly seen in Figure 4 where an arbitrarily chosen eigenstate is given in the two representations. One can see that in the “channel representation”, the eigenstate has a kind of regular and extended structure, while in the “energy representation”, the eigenstate is compressed. In the latter case, one may use a statistical approach to describe the global properties of such eigenstates; see refs. [43,45]. Specifically, these eigenstates can be characterized by introducing an envelope around which the components are expected to fluctuate in a pseudo-random way. We stress that by using this energy ordering, it is possible to relate the global form of eigenstates in the energy representation with its classical counterpart; see, e.g., refs. [29,30,31,32].

## 4. Eigenstates in Energy Representation

In Figure 5, we present two typical pairs of consecutive eigenstates (α=390 and 391 and α=407 and 408). The difference between the eigenstates on the left panels (α=390 and 407) and the other two is clearly qualitative. More specifically, while the states α=390 and 407 are *extended* (in energy) eigenstates, constituted by practically all basis states within the shown energy range, the eigenstates α=391 and 408 are mostly unperturbed: they are extremely *localized* in energy. Indeed, by neglecting all small amplitude components surrounding the main component (see left panels of Figure 5), we can determine the basis state $l_{new}$, defined by the pair (m,n), that most closely resembles the exact eigenstates. We find that this always corresponds to the lowest values of the transversal mode *m*. This fact can be understood by the following physical argument. Consider an eigenstate of the flat billiard $\phi_{m,n}\left(X,Y,k=0\right)\propto \sin\left(m\pi Y/L_{y}\right)\exp\left(iK_{x}X\right)$ with energy $E^{0}=\left(\hbar^{2}/2m_{e}\right)\left(K_{x}^{2}+K_{y}^{2}\right)$, where $K_{y}=m\pi /L_{y}=2\pi /\Lambda_{y}$. Turning on the perturbation (flat to rough billiard) will affect the high energy unperturbed states differently depending mainly on the value of $\Lambda_{y}$. For example, for m=1, the ratio $\Lambda_{y}/W_{1}$ is $2L_{y}/W_{1}\approx 33$ (with $W_{1}=0.06L_{y}$, the value of $W_{1}$ we use throughout this work), which is so large that the state cannot “see” the roughness and thus will remain essentially unperturbed. In contrast, for unperturbed states with the same (or about the same) energy but with large values of *m* (say, $m=62=M_{max}$ and correspondingly small $K_{x}$), their $\Lambda_{y}$ is sufficiently small compared to the amplitude of the roughness ($\Lambda_{y}/W_{1}\approx 0.5$), so that the rough boundary produces a strong mixing of unperturbed levels. The resulting exact eigenstate will consist of many components extended over the energy. Note that we may treat strongly localized states in the channel representation as a kind of scar state.

The fact that the extremely localized (in energy) eigenstates can be identified with the plane waves $\phi_{m,n}\left(x,y\right)$ with small *m*, proper of the flat billiard, makes us expect that they will also be similar to plane waves when presented in configuration representation $|\Psi^{\alpha }{\left(x,y\right)|}^{2}$. Then, in Figure 6 (right), the two localized (in energy) eigenstates of Figure 5 (right) are shown in configuration representation. It is quite unexpected that these eigenstates are very different from the unperturbed ones, even though they are similar in energy representation. Figure 6 (right) shows that the rough boundary “pushes” the probability $|\Psi^{\alpha }{\left(x,y\right)|}^{2}$ away from it. Thus, the eigenstate α=391, whose main component in energy representation is identified with m=1, differs importantly from the unperturbed mode with m=1 whose maximum is at the billiard center $y=L_{y}/2$ (as in the case of the lowest eigenmode of a box of width $L_{y}$ with hard walls). Similar repulsion occurs for the eigenstate α=408 identified with m=2. We stress that this repulsion effect occurs only for rough billiards and is stronger the more rough the billiard boundary is, see, e.g., ref. [39]. A detailed analysis of this repulsion effect will be performed in Section 6. Finally, for comparison purposes, in Figure 6 (left), we also present the two extended (in energy) eigenstates of Figure 5 (left).

In order to characterize quantitatively the eigenstates, we compute various localization measures. The first one is the so-called *entropy localization length* $l_{H}$,
(18)$$l_{H}=\exp\left[-\left(H-H_{\mathrm{GOE}}\right)\right]\approx 2.08\exp\left(-H\right).$$
Here, H stands for the Shannon entropy of an eigenstate in a given basis,
(19)$$H=\sum_{l_{new}=1}^{N}w_{l_{new}}^{\alpha }\;\ln w_{l_{new}}^{\alpha },$$
and $H_{\mathrm{GOE}}$ is the entropy of a completely chaotic state which is characterized by Gaussian fluctuations (for N→∞) of all components $C_{l_{new}}^{\alpha }$ with the same variance $\langle w_{l_{new}}^{\alpha }\rangle =1/N$, where $w_{l_{new}}^{\alpha }={\left|C_{l_{new}}^{\alpha }\right|}^{2}$. The latter property occurs for completely random matrices belonging to a Gaussian Orthogonal Ensemble (GOE). Defined in this way, the quantity $l_{H}$ gives the measure of the effective number of components in an eigenstate. For example, the eigenstates of Figure 5 have $l_{H}\left(\alpha =390\right)=1189.7$, $l_{H}\left(\alpha =391\right)=5.75$, $l_{H}\left(\alpha =407\right)=1517.5$, and $l_{H}\left(\alpha =408\right)=13.16$; that is, extended eigenstates have large values of $l_{H}$, while localized eigenstates are characterized by small values of $l_{H}$.

The second quantity, $l_{ipr}$, which gives another measure of the effective number of components in an eigenstate, is expressed via the *inverse participation ratio*P,
(20)$$l_{ipr}=\left[\frac{P_{\mathrm{GOE}}}{P}\right]\approx \frac{3}{P}$$
with
(21)$$P=\sum_{l=1}^{N}{\left(w_{l_{new}}^{\alpha }\right)}^{2}.$$
where $P_{\mathrm{GOE}}\approx$ 3 is chosen in order to obtain $l_{ipr}=N$ in the GOE limit case. Correspondingly, the eigenstates of Figure 5 have $l_{ipr}\left(\alpha =390\right)=1061.2$, $l_{ipr}\left(\alpha =391\right)=6.32$, $l_{ipr}\left(\alpha =407\right)=1464.4$, and $l_{ipr}\left(\alpha =408\right)=14.99$, where a high correlation with $l_{H}$ can be seen. The above two definitions of localization lengths are the most frequently used when describing the global structure of eigenstates. One should note that these quantities provide an estimate of the effective number of large components, independently on the location of these components, in the unperturbed basis.

To obtain a complete panorama, in Figure 7a,b we plot these two measures, $l_{H}$ and $l_{ipr}$, for the eigenstates |α〉 of the rough billiard. The strong fluctuations of the localization measures are evident in these figures. We can see that neighboring high-energy eigenstates may have drastically different localization measures, which is in agreement with the discussion above about the existence of localized and extended eigenstates. Moreover, these figures give us information about the relative number of each type (localized and extended eigenstates) to be found in a given energy range.

Additional information about the structure of eigenstates can be obtained from the “width” or mean square root $l_{\sigma }$ of an eigenstate, which is computed as
(22)$$l_{\sigma }={\left[\sum_{l_{new}=1}^{N}w_{l_{new}}^{\alpha }\;{\left[l_{new}-n_{c}\left(\alpha \right)\right]}^{2}\right]}^{1/2},$$
where $n_{c}=\sum_{l_{new}}l_{new}\;w_{l_{new}}^{\alpha }$ determines the centroid of an eigenstate in the unperturbed basis.

Comparison of the width $l_{\sigma }$ with $l_{H}$ and $l_{ipr}$ gives the possibility of detecting the so-called *sparsity* of eigenstates. Indeed, small values of the ratio $l_{H}/l_{\sigma }$ (or $l_{ipr}/l_{\sigma }$) indicate that there are “holes” in the structure of the eigenstates; therefore, such eigenstates are *sparse*; see Figure 7c. A detailed analysis shows the existence of sparsed eigenstates. As for the centroid, $n_{c}>\alpha$ observed in Figure 7d indicates that the interaction strength $\hat{V}$ is relatively strong compared to the unperturbed part ${\hat{H}}_{0}$.

The data of Figure 7a,b show the existence of a wide range of values of both localization measures even at high energies; moreover, there are eigenstates with small localization lengths (visible as clusters of points in the lower part of the plots) along the entire energy range. This situation is also present for the same quantities computed for the local density of states (LDOS): that is, for the basis eigenstates $|l_{new}\rangle$ expanded in the exact basis ∣α〉; see Figure 8. Note that to compute the localizations lengths, as well as the centroid, for the LDOS, one must interchange $\alpha ↔l_{new}$ in expressions (Equation 20), (Equation 22) and (Equation 23). It is relevant to stress that the localization lengths for the LDOS show well-defined patterns, see Figure 8, which are not present for the exact eigenstates, see Figure 7. Inspection of Figure 8a–d clearly demonstrates that there is a kind of regularity in the structure of the LDOS: the same type of states appear repeatedly, almost periodically as a function of the basis number $l_{new}$. These figures show the repetition of extremely localized states and of states with different values of the localization measures. The physical origin of all these types of states (localized, intermediate and extended) was explained above, and their appearance can be decoded by examining the structure of the “channel representation” of the Hamiltonian matrix; see Figure 2 (left). A detailed inspection of that matrix shows that the coupling between unperturbed states depends strongly on the values of the index *m*, labeling the transversal modes of the flat billiard. An unperturbed state specified by a large value of *m* (an *m* close to $M_{max}=62$) couples strongly to several other unperturbed states. In contrast, the state with m=1 has practically no coupling to other states. In particular, the extremely localized states, corresponding to the first position on the left line of each brach of Figure 8c occur because of the negligible coupling of the diagonal elements of the $H_{l,l^{\prime }}$ matrix with m=1, the states on the second position of each branch occur for m=3, and so on with *m* odd. Similarly, the states on the right side of the branches result from elements of the $H_{l,l^{\prime }}$ matrix with even values of *m*.

This structure is expected to prevail at all energy ranges, since in any sufficiently large range of energies, there are unperturbed states with all values of $m\in [1,M_{max}]$. Even deep in the semiclassical regime, extremely localized and sparse states will appear but less and less frequently since the energy differences between states of the same type increases with energy.

## 5. Scar-Like States in Energy Space

In this section, we discuss the origin of the scar-like states emerging in the energy representation; they seem to be generic for plane billiards with rough surfaces. The starting point is the observation that for a weak roughness, these states can be associated with those unperturbed states that have a small value of the index *m*, especially for m=1. Since the localized eigenstates identified above with m=1 exist at all energies, the important question is about the fraction of these states relative to the total number of states $N\left(E\right)\equiv \sum_{i}\left\{i|E_{i}\leq E\right\}$ as a function of energy. This question is important in view of Shnirel’man’s theorem [14] stating that the eigenfunctions of a classically ergodic system are equidistributed over the energy shell in the classical limit. In other words, we expect that
(23)$$\lim_{E\to \infty }\frac{N_{m=1}\left(E\right)}{N(E)}=0,$$
where $N_{m=1}\left(E\right)$ is the number of m=1 eigenstates up to some energy *E*:(24)$$N_{m=1}\left(E\right)\equiv \sum_{i}\left\{i|E_{i}\leq E;\Psi_{i}≅\phi_{1n}\left(u,v\right)\right\}.$$
Neither Shnirel’man’s theorem nor Equation (Equation 24) says anything about how the limit is approached. Moreover, Shnirel’man’s theorem does not consider the possible existence of parabolic fixed points. These occur in our billiard; they are the bouncing ball orbits or the continuous set of all horizontal trajectories ($P_{y}=0$) which do not hit the two boundaries. Quantum mechanically, there are no $P_{y}=0$ states, but the eigenstates of the flat channel with a minimum value of $P_{y}$ are the m=1 states. Thus, we have referred to the m=1 eigenstates as “bouncing ball states” in [38].

For $W_{1}/L_{y}\ll 1$, one can use the unperturbed spectra given by Equation (Equation 16) to obtain a good estimate for $N_{m=1}\left(E\right)$ and N(E). For large $N_{max}$ and $M_{max}$, Equation (Equation 16) represents, ignoring the $k^{2}$ term, half of the ellipse
(25)$$\frac{n^{2}}{\frac{2m_{e}}{\hbar^{2}}{\left(\frac{L_{x}}{2\pi }\right)}^{2}E}+\frac{m^{2}}{\frac{2m_{e}}{\hbar^{2}}{\left(\frac{L_{y}}{\pi }\right)}^{2}E}=1,$$
in the n−m plane (recall that $m\in [1,M_{max}]$ and $n\in [-N_{max},N_{max}]$); above $E\equiv E_{n,m}^{\left(0\right)}\left(k\right)≅E_{n,m}^{\left(0\right)}\left(0\right)$. In Figure 9, we show the n−m plane for the case of $N_{max}=32$ and $M_{max}=62$; there, the dots represent basis states and the red ellipse is Equation (Equation 26) with $E=400(\hbar^{2}/2m_{e})$ and $L_{x}=Ly=2\pi$. Then, for large *E*, the number N(E) equals the area of the half ellipse, and the number $N_{m=1}\left(E\right)$ of localized eigenstates equals twice the size or its minor axis:(26)$$N\left(E\right)=\left(\frac{2m_{e}}{\hbar^{2}}\right)\frac{\sigma }{4\pi }E,$$
(27)$$N_{m=1}={\left(\frac{2m_{e}}{\hbar^{2}}\right)}^{1/2}\frac{\sigma }{\pi }\frac{\sqrt{E}}{L_{y}};$$
therefore,
(28)$$\frac{N_{m=1}\left(E\right)}{N(E)}={\left(\frac{\hbar^{2}}{2m_{e}}\right)}^{1/2}\frac{4L_{x}}{\sigma \sqrt{E}},$$
where $\sigma =L_{x}L_{y}$ is the area of one period of the billiard.

We see that Equation (Equation 27) is precisely the first term in the Weyl series for the integrated density of states (see for example [46], Section 7). Equation (Equation 29) indicates that (i) for fixed *E* and σ, the portion of m=1 states, relative to the total number of states, is larger the narrower the billiard is and (ii) the convergence of $N_{m=1}\left(E\right)/N\left(E\right)$ to zero is rather slow; it decreases as a power-law of *E*.

It is important to stress that in the case of a rough billiard having a highly modulated boundary composed of many harmonics, the expression for N(E) is not simply given by (Equation 27), since now, the perimeter γ of the boundary, as well as its curvature, contribute significantly to N(E) as [47]
(29)$$\begin{array}{c} N\left(E\right)=\left(\frac{2m_{e}}{\hbar^{2}}\right)\frac{\sigma }{4\pi }E-\frac{\gamma }{4\pi }{\left(\frac{2m_{e}}{\hbar^{2}}\right)}^{1/2}\sqrt{E}-{\left(\frac{2m_{e}}{\hbar^{2}}\right)}^{1/2}\frac{\sqrt{E}}{\pi }\sum_{r=1}^{\infty }\frac{{\left(-1\right)}^{r}}{\left(r-1/2\right)E^{r}}C_{2r+1}. \end{array}$$
The coefficients $C_{2r+1}$, r≥1, in the sum of the Weyl series (Equation 30), depend on the curvature (and its derivative) of the modulated boundary (see 3 of [47]). As is shown in [47], the higher the index *r* of the Weyl coefficient $C_{2r+1}$, the more complex the expansions for $C_{r}$ are, involving higher and higher powers of the curvature and its derivatives. Clearly, the larger the number of harmonics $N_{T}$, the more important the perimeter and curvature terms become.

In addition, it is also instructive to look at the location of the eigenstates on the n−m plane. Thus, in Figure 10, we present the eigenstates of Figure 5 also in energy representation but now on the n−m plane. From this figure, we can observe that: (i) the components of the energy-extended eigenstates of Figure 5 (left) are not equidistributed on the n−m plane; instead, their main components are concentrated around the region with n∼0 and large *m*. That the components are concentrated around n∼0 supports the fact that these eigenstates are not extended over the *x*-axis when plotted in configuration representation; see Figure 6 (left). (ii) The components of the energy-localized eigenstates of Figure 5 (right) are also localized on the n−m plane; they are indeed characterized by a single value of *n* and several but small values of *m* (a detailed decoding of this *m* dependence will be performed in the next section).

## 6. Scars and Billiard Symmetries; Repulsion Effect

In this section, we study in detail the repulsion, in configuration representation, that suffer the energy-localized eigenstates of rough billiards from the rough boundaries. This effect, already shown in Figure 6right panels, was first reported in [38,39] but also observed in [24,48,49]. Moreover, in addition to the one-flat-boundary billiard of Figure 11a, we now consider two additional billiard geometries: the antisymmetric billiard, Figure 11b, and the symmetric billiard, Figure 11c; see also [24,25].

We start by presenting in Figure 12 few typical extremely energy-localized eigenstates for the billiards of Figure 11 in three different energy regions. The main component of all those eigenstates can be identified with an unperturbed state with m=1 (see also Figure 13) where we present again the eigenstates of Figure 12 but now in configuration representation). Close inspection of Figure 12 shows that for the one-flat-boundary billiard and the symmetric billiard, panels (a) and (c) of Figure 12, respectively, the localized eigenstates have several components of appreciable magnitude. This is in contrast with the eigenstates of the antisymmetric billiard, see panel (b) of Figure 12, which has only one effective component with magnitude very close to unity.

The decoding of the main components of the energy-localized eigenstates of the one–flat–boundary billiard, corresponding to panels (a) of Figure 12, shows that the second most important component of each of the localized states corresponds to m=2, the next one to m=3 and so on, all with the same value of *n*. That is, there is no mixing between different values of *n*. Moreover, the amplitudes of these eigenstate components decay exponentially as a function of *m*; see Figure 14 (left).

On the other hand, by decoding the main components of the energy-localized eigenstates of the symmetric billiard, corresponding to panels (c) of Figure 12, we see that the second most important component of each of the localized states corresponds to m=3, the next one to m=5 and so on, all with the same value of *n* but with alternating signs. Again, there is no mixing between different values of *n*. In addition, the amplitudes of these eigenstate components show an exponential decay as a function of *m*, which is in this case modulated by a sinus function; see Figure 14 (right).

Following the above observations, one can substitute in Equation (Equation 14) the following expressions: (30)$$\begin{array}{ccc} C_{mn}^{\alpha }\left[a\right] & ≅ & S_{a}\exp\left[-\beta_{a}\left(m-1\right)\right]\delta_{nn_{\alpha }}\;, \end{array}$$(31)$$\begin{array}{ccc} C_{mn}^{\alpha }\left[b\right] & ≅ & \delta_{m,1}\delta_{nn_{\alpha }}\;, \end{array}$$(32)$$\begin{array}{ccc} C_{mn}^{\alpha }\left[c\right] & ≅ & S_{c}\sin\left(m\pi /2\right)\exp\left[-\beta_{c}\left(m-1\right)\right]\delta_{nn_{\alpha }}\;, \end{array}$$
for the energy-localized eigenstates of [*a*] the one–flat–boundary billiard, [*b*] the antisymmetric billiard, and [*c*] the symmetric billiard, respectively. Here, $n_{\alpha }$ corresponds to the value of *n* characterizing all the main components of a given energy-localized eigenstate.

The parameters $S_{a}=0.8$, $\beta_{a}=0.53$, $S_{c}=0.9$, and $\beta_{c}=0.426$ are obtained by fitting the data to the dependence given by the expressions for $C_{mn}^{\alpha }$ above; see the fittings in Figure 14. It is important to remark that for a fixed $N_{T}$, all energy-localized states are characterized by the same values of $S_{a,b}$ and $\beta_{a,b}$. Substituting the dependence of $C_{mn}^{\alpha }$ into Equation (Equation 14) gives
$$\begin{array}{ccc} \Psi_{loc}^{\alpha }\left(u,v\right)\left[a\right] & ≅ & C_{a}S_{a}\frac{\exp\left[i(k+n_{\alpha })u\right]}{\pi^{1/2}g^{1/4}}\sum_{m=1}^{\infty }\sin\left(\frac{m\pi v}{L_{y}}\right)\exp\left[-\beta (m-1)\right]\;,\\ \Psi_{loc}^{\alpha }\left(u,v\right)\left[b\right] & ≅ & C_{c}\frac{\exp\left[i(k+n_{\alpha })u\right]}{\pi^{1/2}g^{1/4}}\sin\left(\frac{\pi v}{L_{y}}\right)\;,\\ \Psi_{loc}^{\alpha }\left(u,v\right)\left[c\right] & ≅ & C_{c}S_{c}\frac{\exp\left[i(k+n_{\alpha })u\right]}{\pi^{1/2}g^{1/4}}\sum_{m=1}^{\infty }\sin\left(\frac{m\pi }{2}\right)\sin\left(\frac{m\pi v}{L_{y}}\right)\exp\left[-\beta (m-1)\right]\;, \end{array}$$
where $C_{a,b,c}$ arise to satisfy the orthonormality condition in curvilinear coordinates,
(33)$$\int_{0}^{L_{y}}\int_{0}^{L_{x}}dudv\sqrt{g}\Psi_{loc}^{\alpha \;*}\Psi_{loc}^{\alpha }=1\;.$$

Then, it follows that
$$\begin{array}{ccc} & & |\Psi_{loc}^{\alpha }\left(y\right){|}^{2}\left[a\right]≅\frac{1}{\pi }{\left(\sum_{m=1}^{\infty }\exp\left[-2\beta_{a}\left(m-1\right)\right]\right)}^{-1}{\left[\sum_{m=1}^{\infty }\sin\left(\frac{m\pi y}{L_{y}}\right)\exp\left[-\beta_{a}\left(m-1\right)\right]\right]}^{2}\;,\\ & & |\Psi_{loc}^{\alpha }\left(y\right){|}^{2}\left[b\right]≅\frac{1}{\pi }\sin\left(\frac{\pi y}{L_{y}}\right)\;,\\ & & |\Psi_{loc}^{\alpha }\left(y\right){|}^{2}\left[c\right]≅\\ & & \;\;\frac{1}{\pi }{\left(\sum_{m=1}^{\infty }\sin^{2}\left(\frac{m\pi }{2}\right)\exp\left[-2\beta_{c}\left(m-1\right)\right]\right)}^{-1}{\left[\sum_{m=1}^{\infty }\sin\left(\frac{m\pi }{2}\right)\sin\left(\frac{m\pi y}{L_{y}}\right)\exp\left[-\beta_{c}\left(m-1\right)\right]\right]}^{2}. \end{array}$$

These latter expressions give the average shape of the localized eigenstates in the configuration representation, projected onto the $y-|\Psi^{\alpha }{|}^{2}$ plane. In addition, note that $\Psi_{loc}^{\alpha }\left(y\right)$ does not depend on the parameters $S_{a,c}$, so the relevant parameters are $\beta_{a,b}$. To obtain $|\Psi_{loc}^{\alpha }\left(y\right){|}^{2}$ above, we have made the approximation v≈y and g≈1 since $W_{1}/L_{y}=0.06\ll 1$.

Finally, after some algebra, we obtain
(34)$$\begin{array}{ccc} |\Psi_{loc}^{\alpha }\left(y\right){|}^{2}\left[a\right] & ≅ & \frac{\left[\exp\left(2\beta_{a}\right)-1\right]\sin^{2}\left(\pi y/L_{y}\right)}{4\pi {\left[\cos\left(\pi y/L_{y}\right)-\cosh\left(\beta_{a}\right)\right]}^{2}}\;, \end{array}$$
(35)$$\begin{array}{ccc} |\Psi_{loc}^{\alpha }\left(y\right){|}^{2}\left[b\right] & ≅ & \frac{1}{\pi }\sin\left(\frac{\pi y}{L_{y}}\right)\;, \end{array}$$
(36)$$\begin{array}{ccc} |\Psi_{loc}^{\alpha }\left(y\right){|}^{2}\left[c\right] & ≅ & \frac{\sinh\left(2\beta_{c}\right)\sinh^{2}\left(\beta_{c}\right)\sin^{2}\left(\pi y/L_{y}\right)}{2\pi {\left[\cosh^{2}\left(\beta_{c}\right)-\sin^{2}\left(\pi y/L_{y}\right)\right]}^{2}}. \end{array}$$
Note that when $\beta_{a,b}\to \infty$, i.e., when the energy-localized eigenstates have only one single component, $|\Psi_{loc}^{\alpha }\left(y\right){|}^{2}\left[a\right]=|\Psi_{loc}^{\alpha }\left(y\right){|}^{2}\left[c\right]=\left(1/\pi \right)\sin\left(\pi y/L_{y}\right)$, as required.

In Figure 15, we plot Equations (Equation 35)–(Equation 37) together with the numerical data from the eigenstates of Figure 13. From this figure, it is clear that (i) in the one-flat-boundary billiard, the energy-localized eigenstates are repelled from the rough boundary toward the flat boundary; see the curves labeled with (a), as already observed in Figure 6; (ii) in the symmetric billiard, the energy-localized eigenstates are repelled from both rough boundaries toward the billiard center; see the curves labeled with (c); while (iii) in the antisymmetric billiard, the repulsion effect is completely absent, so the eigenstates correspond to those of the flat billiard; see the curves labeled with (b). A detailed study [24] revealed that the absence of repulsion in the antisymmetric billiard is due to the fact that in this billiard, there is no square-gradient scattering between different modes or channels, in contrast to the one-flat-boundary billiard and the symmetric billiard. Thus, one can conclude that the main contribution to the repulsion is due to the intermode square-gradient scattering terms of the Hamiltonian matrix (i.e., the integrals $J_{3}$, $J_{6}$, and $J_{7}$ in Equation (Equation 18), which vanish for the antisymmetric billiard). This conclusion supports the observation made above according to which the effect of repulsion can be explained as due to a strong localization in the channel space. This localization occurs due to a relatively strong interaction between different conducting channels (or billiard modes).

Note that since the parameters $S_{a,b}$ and $\beta_{a,b}$ are the same for all m=1 eigenstates for a fixed $N_{T}$, one can find $S_{a,b}$ and $\beta_{a,b}$ for some low-energy eigenstate (so that the matrix to be diagonalized is small) and with this one can infer the global shape of all m=1 eigenstates. Conversely, knowing the shape of an m=1 eigenstate within any energy range, one may assess by looking at the amount of repulsion the degree of complexity, presumably unknown, of the boundaries.

The excellent agreement between Equations (Equation 35)–(Equation 37) and the numerical data from the eigenstates of Figure 13 indicates that for extremely localized eigenstates with m=1, there is no need to diagonalize the whole matrix since, there is no mixing between different values of *n*. In order to obtain a good approximation of these eigenstates, one only needs to diagonalize the block of size $M_{max}\times M_{max}$ corresponding to the quantum number $n_{\alpha }$, which is a good quantum number in the presence of perturbation.

Within this approach, it is also possible to study energy-localized eigenstates with m>1 using blocks of the Hamiltonian (Equation 18) of size $M_{max}\times M_{max}$. So, in Figure 16, we plot the 2nd, 3rd, and 4th lowest eigenstates in configuration representation of a block of the Hamiltonian (Equation 18) of size 100×100 for the one-flat-boundary billiard (left panels), an antisymmetric billiard (central panels), and the symmetric billiard (right panels). They correspond to eigenstates characterized by m=2, 3, and 4, respectively. As expected, they have 2, 3, and 4 maxima in the transverse direction, respectively. As well as for the eigenstates characterized by m=1, see Figure 13 and Figure 15, the eigenstates of Figure 16 suffer repulsion from the rough boundaries in the one-flat-boundary billiard and the symmetric billiard, while in the antisymmetric billiard, the repulsion is absent.

## 7. Conclusions

In this paper, we have studied the properties of rectangular billiards with one and two rough boundaries. In the classical description, such billiards can be considered as completely chaotic, although this has not been rigorously proven. Our interest was to understand the typical structure of the eigenfunctions in the quantum description and to discover the mechanism of occurrence of scar states. For numerical simulation, we have chosen the method of reducing the original model with one particle moving inside the billiard to a model of two artificial particles in a billiard with flat boundaries, however with an effective interaction between the particles. Thus, the complexity of the rough boundary is embedded into a quite complicated interaction potential. In this representation, the Hamiltonian of the system can naturally be represented as the sum of two terms, one of which corresponds to the motion of two particles in a billiard with flat walls, and the second term describes the interaction between particles. As can be seen, such a scheme is similar to that which is often used in physics when describing isolated systems with many interacting particles. Thus, the question arises about the correspondence between the properties of eigenfunctions in a many-particle basis (in our case, a two-particle basis) and the properties of eigenfunctions in the configuration space. As was found, this correspondence is not simple, especially if we are interested in the degree of localization of the eigenfunctions and, in particular, in the occurrence of scar states.

As a result of our study, we came to the conclusion that one can roughly speak of three different types of eigenstates. The first type includes states whose density in the configuration basis is concentrated in the vicinity of bouncing ball trajectories that are perpendicular to two horizontal surfaces. Such highly localized functions (with some degree of chaos) are analogous to well-studied scar states in stadium billiards. The second type of eigenfunctions, which are strongly localized in the momentum space, can be associated with the plane waves in the billiard with flat boundaries. To some extent, they can also be termed as scar states because of their strong correspondence to unperturbed eigenstates. There is a third type of eigenfunctions that consists of many components in both the energy space of the Hamiltonian and in the configuration space. These eigenstates are quite complicated and can be treated as partially chaotic. Thus, it can be assumed that these eigenstates quickly become completely chaotic with increasing energy in comparison with those functions that are strongly localized either in energy or in configuration space.

We have considered in detail how the number of eigenstates strongly localized in the energy space decreases with increasing energy. This question is related to Shnirel’man’s theorem, according to which, in the classical limit, the measure (number) of scar functions decreases to zero. It is worth noting that this theorem does not predict at what rate (in energy) all eigenfunctions become ergodic. In the general case of ergodic billiards, this question remains open, but in our case, the estimate indicates that the number of scar states (in the energy space) decreases rather slowly, namely, inversely proportional to the square of the energy.

We were also interested in the “repulsion” effect, discovered in [22], according to which for a billiard with one corrugated surface, some of the eigenstates suffer a shift of the maximal density toward the surface, which is flat. Our numerical data showed that this effect is highly pronounced for those eigenfunctions that are close to plane waves traveling in the horizontal direction. For a subset of such eigenstates, we have derived an approximate expression that clearly indicates an exponential localization in the momentum space. Even more interesting was to compare the structure of this type of eigenstate for the billiard with two corrugated surfaces. We have found that for symmetric rough surfaces, the repulsion effect is strongly enhanced. On the other hand, for antisymmetric profiles, the repulsion is absent. The origin of this phenomenon can be explained by analyzing the structure of the Hamiltonian matrix, which is strongly influenced by the form of the off-diagonal matrix elements that depend on the amplitude, first derivative, and the second derivative of the billiard profiles. Apart from that, there are terms that describe the inter-correlations between profiles. Our analysis explains both the repulsion enhancement and the disappearance of repulsion. Due to this analysis, one can predict the impact of the type of symmetry between the profiles on the scattering properties of quasi-one-dimensional waveguides with corrugated surfaces.

This article is dedicated to Professor Giulio Casati on the occasion of his 80th birthday. We wish him to be healthy and scientifically productive for the next 20 years.

## Figures and Tables

**Figure 1 entropy-25-00189-f001:**
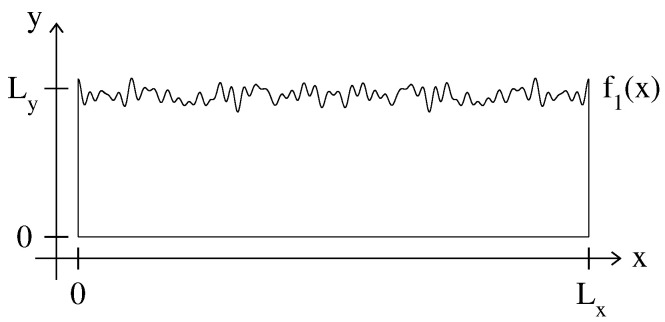
Example geometry for the rough billiard. $f_{1}\left(x\right)=L_{y}+W_{1}\xi_{1}\left(x\right)$ and $f_{2}\left(x\right)=0$; see Equation (Equation 1). Here, $\xi_{1}\left(x\right)$ is given by Equation (Equation 2) with $N_{T}=30$ and $W_{1}=L_{y}/10$. The harmonics $A_{k}$ used for $\xi_{1}\left(x\right)$ are drawn from a flat random distribution defined in the interval [−A,A] (with A such that $|\xi_{1}\left(x\right)|\leq 1$).

**Figure 2 entropy-25-00189-f002:**
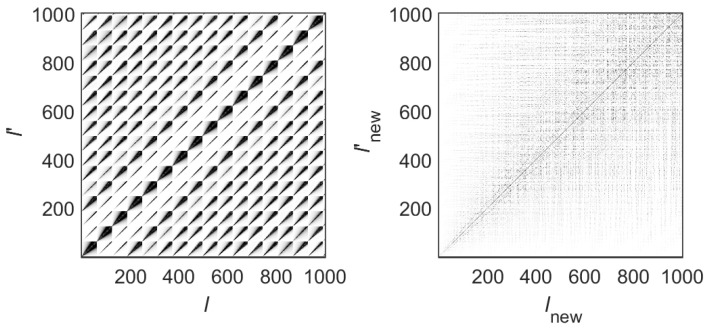
(**left**) Lower part of the Hamiltonian matrix in the “channel representation” $|H_{l,l^{\prime }}|$. (**right**) Lower part of the Hamiltonian matrix in the “energy representation” $|H_{l_{new},l_{new}^{\prime }}|$. $L_{x}=2\pi$, $L_{y}=2\pi$, $W_{1}=0.06L_{y}$, k=0.1, $N_{T}=100$, and $\left(N_{max},M_{max}\right)=\left(32,62\right)$ were used. With this choice of $N_{max}$ and $M_{max}$, we obtain $L_{max}=4030$. Both matrices are shown in grayscale where darker means higher amplitude of the matrix element.

**Figure 3 entropy-25-00189-f003:**
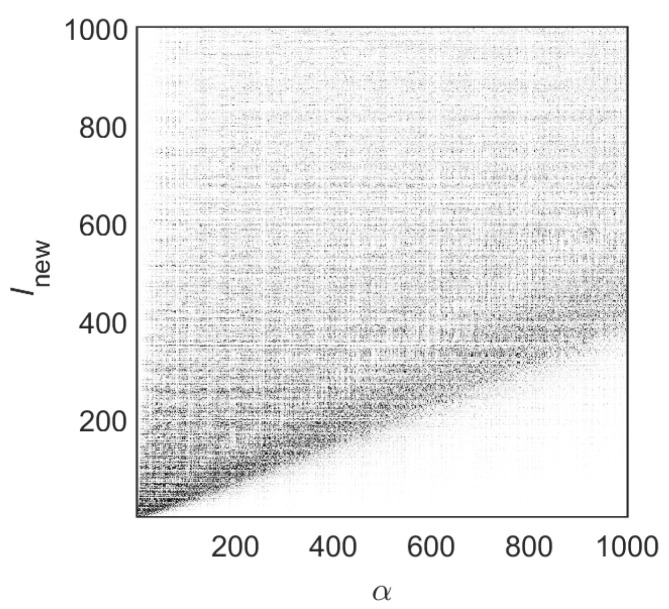
Lower part of the state matrix $|C_{l_{new}}^{\alpha }{|}^{2}$ from the Hamiltonian matrix of Figure 2 (right); that is, $L_{x}=2\pi$, $L_{y}=2\pi$, $W_{1}=0.06L_{y}$, k=0.1, $N_{T}=100$, and $\left(N_{max},M_{max}\right)=\left(32,62\right)$ were used. The matrix is shown in grayscale where darker means higher amplitude of the matrix element.

**Figure 4 entropy-25-00189-f004:**
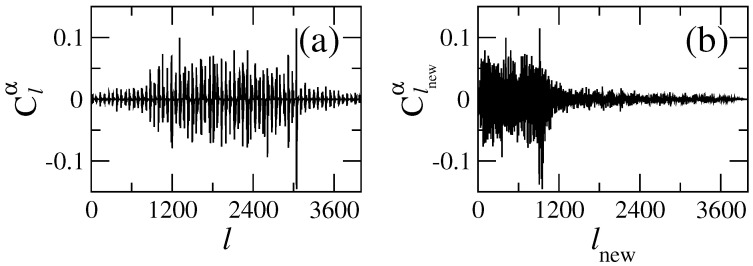
Example of an eigenstate obtained from the Hamiltonian in (**a**) the “channel representation” and (**b**) the “energy representation”. $L_{x}=2\pi$, $L_{y}=2\pi$, $W_{1}=0.06L_{y}$, k=0.1, and $N_{T}=100$ were used. (**a**) The eigenstate α=392 as a function of *l* and (**b**) the same eigenstate as a function of $l_{new}$.

**Figure 5 entropy-25-00189-f005:**
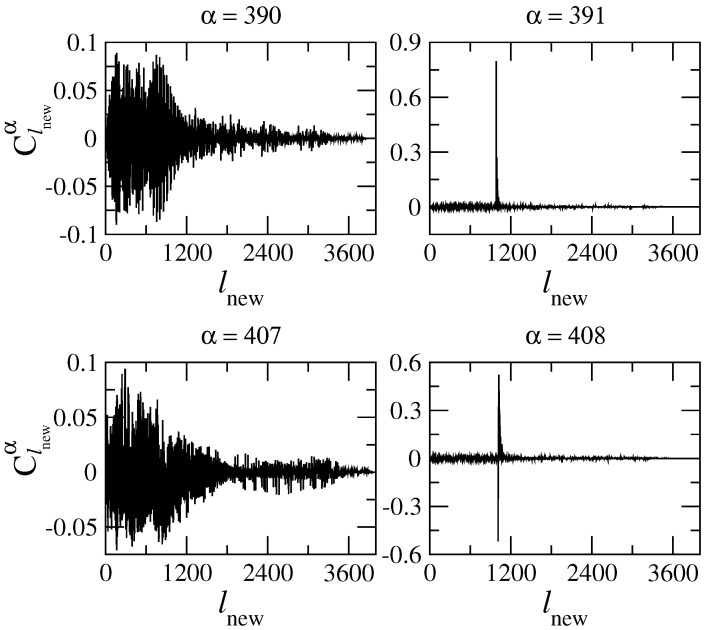
Typical pairs of consecutive eigenstates (α=390, 391, 407 and 408) in the energy representation for $L_{x}=2\pi$, $L_{y}=2\pi$, $W_{1}=0.06L_{y}$, k=0.1, and $N_{T}=100$.

**Figure 6 entropy-25-00189-f006:**
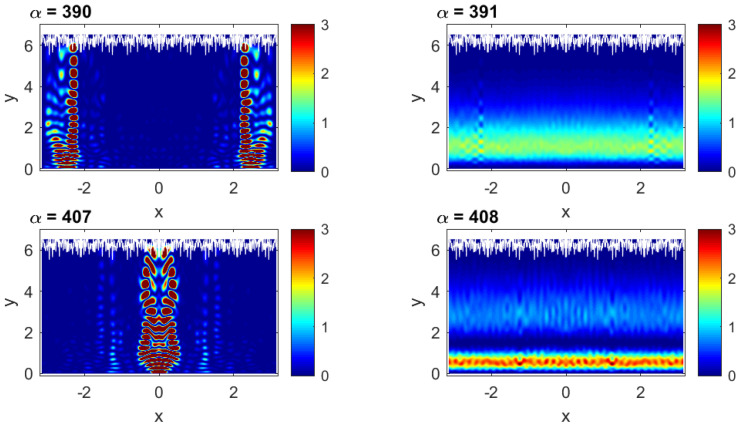
The eigenstates of Figure 5 in the configuration representation $|\Psi^{\alpha }{\left(x,y\right)|}^{2}$. The scale of the color code on the right of the panels should be multiplied by $10^{-3}$.

**Figure 7 entropy-25-00189-f007:**
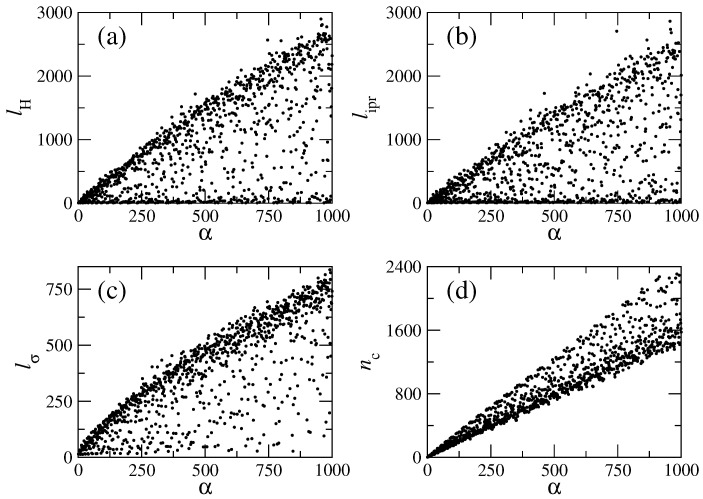
Localization measures for the eigenstates α of the rough billiard. (**a**) Entropy localization length $l_{H}$, (**b**) inverse participation ratio $l_{ipr}$, (**c**) mean square root $l_{\sigma }$, and (**d**) centroid $n_{c}$. Here, $L_{x}=2\pi$, $L_{y}=2\pi$, $W_{1}=0.06L_{y}$, k=0.1, and $N_{T}=100$ were used.

**Figure 8 entropy-25-00189-f008:**
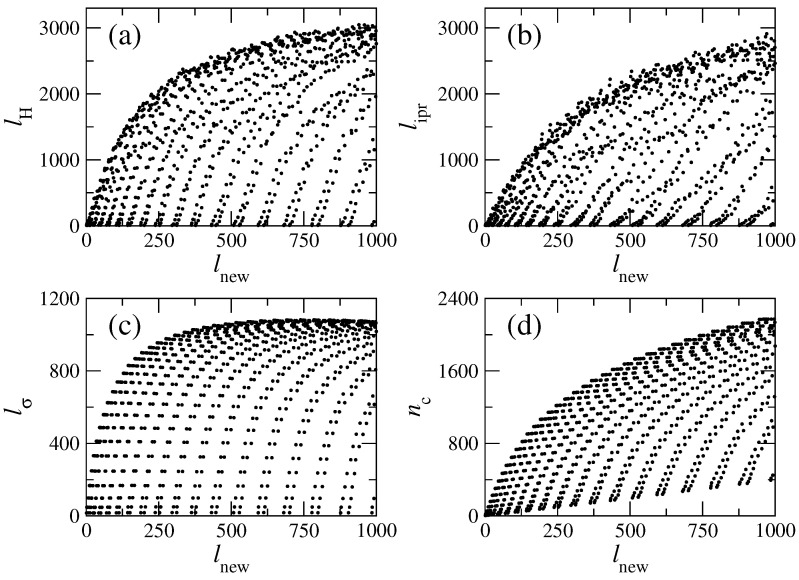
Localization measures for the LDOS, $l_{new}$, of the rough billiard. (**a**) $l_{H}$, (**b**) $l_{ipr}$, (**c**) $l_{\sigma }$, and (**d**) $n_{c}$. $L_{x}=2\pi$, $L_{y}=2\pi$, $W_{1}=0.06L_{y}$, k=0.1, and $N_{T}=100$ were used.

**Figure 9 entropy-25-00189-f009:**
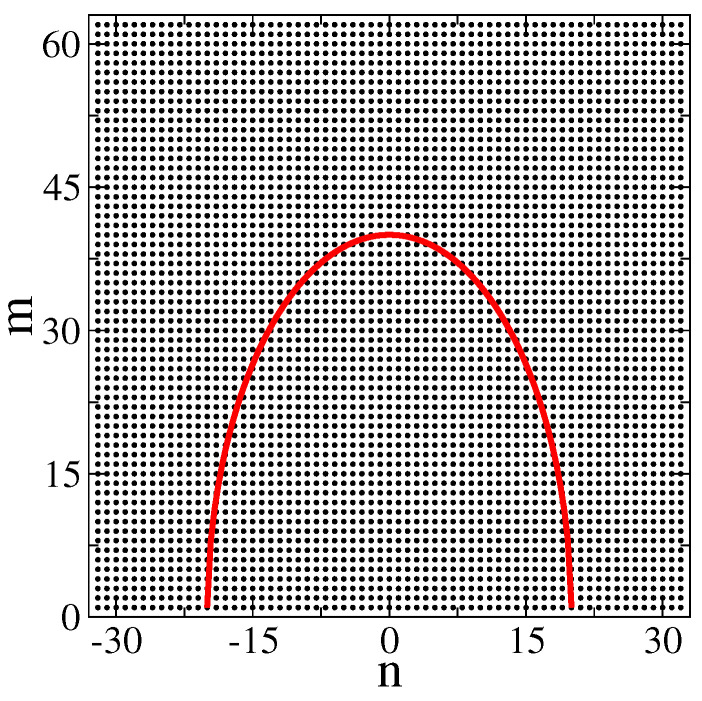
The n−m plane for the case of $N_{max}=32$ and $M_{max}=62$. Dots represent basis states ∣l〉≡∣m,n〉 or $|l_{new}\rangle \equiv |m,n\rangle$. The red ellipse is Equation (Equation 26) with $E=400(\hbar^{2}/2m_{e})$ and $L_{x}=Ly=2\pi$.

**Figure 10 entropy-25-00189-f010:**
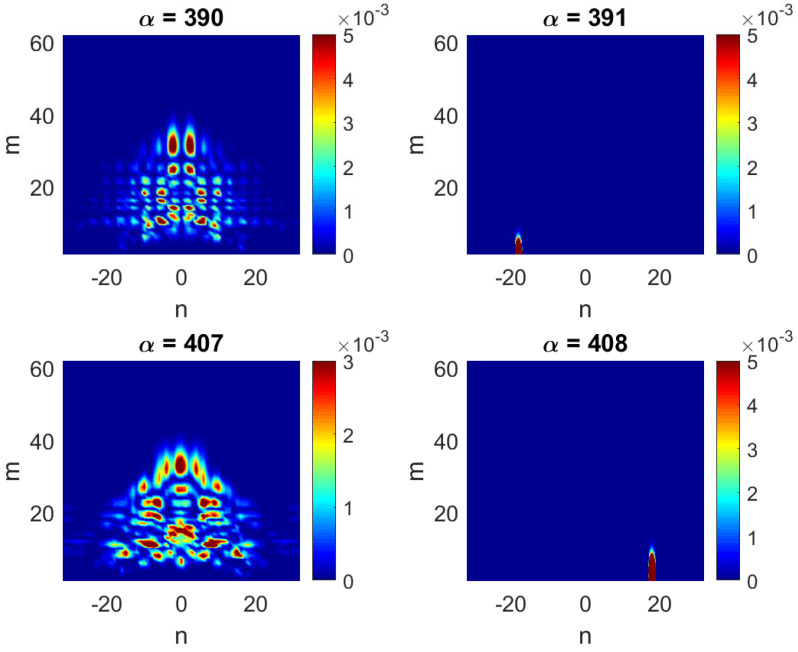
The eigenstates α=390, 391, 407 and 408 in energy representation (see Figure 5) on the n−m plane.

**Figure 11 entropy-25-00189-f011:**
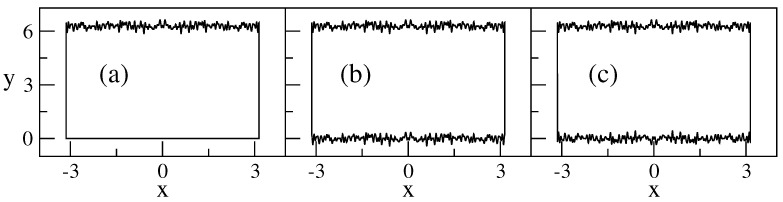
The rough billiards analyzed below: (**a**) a billiard with one flat boundary, $W_{2}=0$; (**b**) an antisymmetric billiard, $W_{1}=W_{2}$; and (**c**) a symmetric billiard $W_{1}=-W_{2}$. $N_{T}=100$, $W_{1}/L_{y}=0.06$, $L_{x}/L_{y}=1$, $L_{x}=2\pi$, and $\xi_{1}\left(x\right)=\xi_{2}\left(x\right)$.

**Figure 12 entropy-25-00189-f012:**
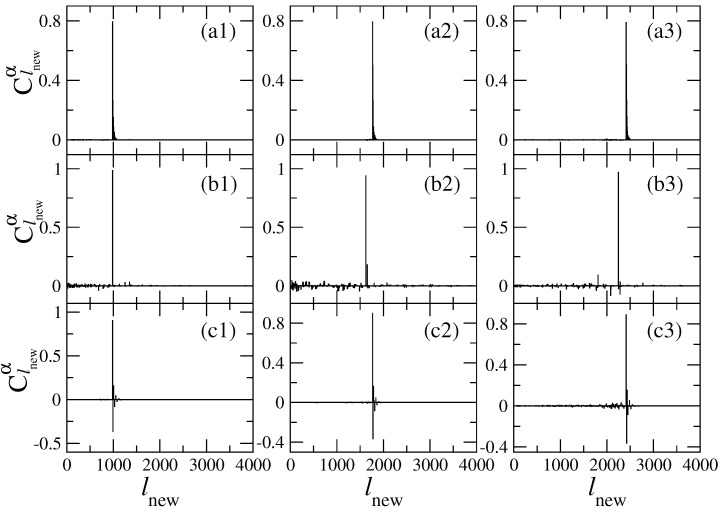
Localized eigenstates in the energy representation $C_{l_{new}}^{\alpha }$ for (**a**) a one-flat-boundary billiard, (**b**) an antisymmetric billiard, and (**c**) a symmetric billiard; see Figure 11. Specifically, (**a1**) α=391, (**a2**) α=705, (**a3**) α=968, (**b1**) α=170, (**b2**) α=283, (**b3**) α=401, (**c1**) α=389, (**c2**) α=704, and (**c3**) α=962. The main components of these localized eigenstates in energy representation correspond to unperturbed states characterized by m=1 and (**a1**) n=−18, (**a2**) n=−24, (**a3**) n=−28, (**b1**) n=−18, (**b2**) n=−23, (**b3**) n=−27, (**c1**) n=−18, (**c2**) n=−24, and (**c3**) n=−28. $N_{T}=100$, $W_{1}/L_{y}=0.06$, $L_{x}/L_{y}=1$, $L_{x}=2\pi$, k=0.1, and $\xi_{1}\left(x\right)=\xi_{2}\left(x\right)$ were used.

**Figure 13 entropy-25-00189-f013:**
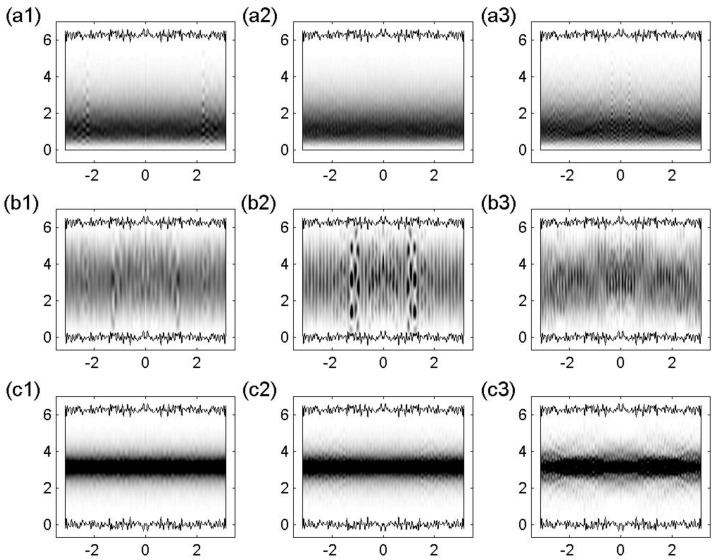
Localized eigenstates of Figure 12 in the configuration representation $|\Psi^{\alpha }{\left(x,y\right)|}^{2}$. Eigenstates for (**a1**–**a3**) a one-flat-boundary billiard, (**b1**–**b3**) an antisymmetric billiard, and (**c1**–**c3**) a symmetric billiard; same labeling as in Figure 12.

**Figure 14 entropy-25-00189-f014:**
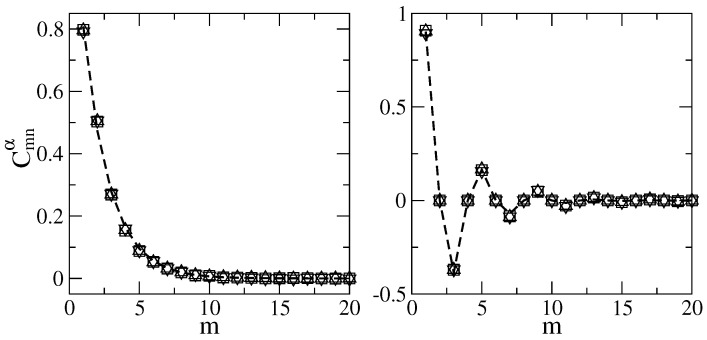
Main components $C_{mn}^{\alpha }$ as the functions of *m* from the localized eigenstates in the energy representation of Figure 13a (right) and Figure 13c (left). Different symbols correspond to different eigenstates. The thick dashed curves are the best fits to Equation (Equation 31) (right) and Equation (32) (left) with $S_{a}=0.8$ and $\beta_{a}=0.53$ and $S_{c}=0.9$ and $\beta_{c}=0.426$, respectively.

**Figure 15 entropy-25-00189-f015:**
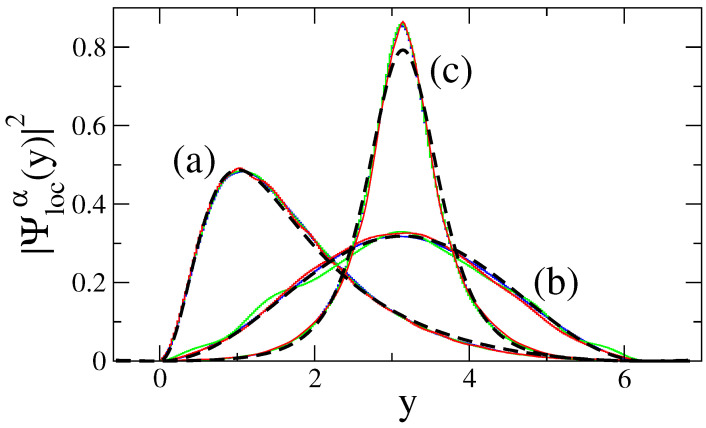
Projection of the eigenstate profiles of Figure 13 (color curves) onto the *y* coordinate together with the analytical expressions for $|\Psi_{loc}^{\alpha }\left(y\right){|}^{2}$ (thick dashed lines) given by Equations (Equation 35)–(Equation 37) with $\beta_{a}=0.53$ and $\beta_{c}=0.426$. Profiles of eigenstates for (**a**) a one-flat-boundary billiard, (**b**) an antisymmetric billiard, and (**c**) a symmetric billiard.

**Figure 16 entropy-25-00189-f016:**
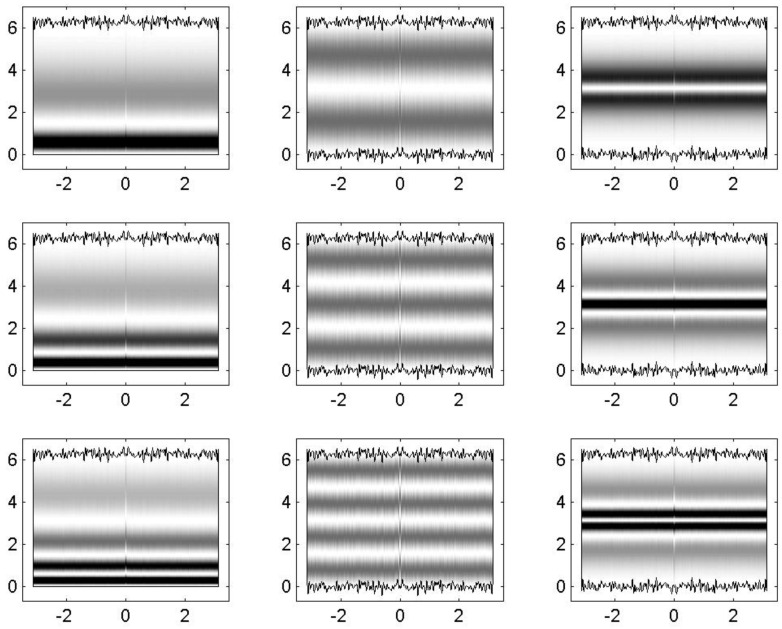
Left column: 2*^nd^*, 3*^rd^*, and 4*^rd^* lowest eigenstates in the configuration representation $|\Psi^{\alpha }{\left(x,y\right)|}^{2}$ for a block of the Hamiltonian (Equation 18) of size $M_{max}\times M_{max}$ for the billiards of Figure 11a. Middle and right column: the same but for the billiards of Figure 11b,c, respectively. $N_{T}=100$, $W_{1}/L_{y}=0.06$, $L_{x}/L_{y}=1$, $L_{x}=2\pi$, $\xi_{1}\left(x\right)=\xi_{2}\left(x\right)$, $M_{max}=100$, $n_{\alpha }=0$, and k=0 were used.

## Data Availability

The data presented in this study are available on request from the corresponding author.

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
