# Peer review of "Scarring in Rough Rectangular Billiards"

_entropy, 2023, doi:10.3390/e25020189_

Round 1
Reviewer 1 Report
The manuscript presents a study of scarring of eigenstates in rectangular billiards with slightly corrugated surface. It is shown that there are two sets of scarred states, one related to the bouncing ball trajectories in the configuration space and a second one, originated from the plane-wave states of the unperturbed flat billiard. The analysis is done employing a two-particle basis, where the roughness of the billiard boundaries is absorbed in a complicated potential. It is numerically shown that in billiards with one rough surface exists repulsion of eigenstates from this surface. When two horizontal rough surfaces are considered, the repulsion effect is either enhanced or canceled depending on whether the rough profiles are symmetric or antisymmetric.
The present study builds on the previous works from some of the authors, Refs. 20-23 and 36-37, where the model was presented and discussed in detail. The present contribution is a valuable application of the previous work, centered in the analysis of scarred states, showing how a specific definition and classification of the unperturbed basis allows for a clear identification of extended and localized states in this basis, providing analytical expressions for the wave functions of the energy-localized states in this basis. Attention is paid to the way in which the number of eigenstates strongly localized in the energy space slowly decreases with increasing energy.
I recommend the publication of the manuscript, after the following minor suggestions are taken into account:
Page 2: The phrases “It can be seen that in this case the eigenfunctions are not only ergodic but also maximally chaotic (random).”
and “the typical mechanism for scars is the presence of local symmetries of the system, which can correspond to the presence of local integrals of motion.”
beg for explanations, or at least references.
Page 3. “horizontal” and “vertical” trajectories are mentioned before the model is introduced. Perhaps “parallel” and “perpendicular” to the walls would be clearer.
Page 3. In Eq. (2) the coefficients A_k are introduced, but along the article there is no mention as how they are selected. This information is clearly missing in the caption of Fig. 1.
Page 8. In Figs 2 and 3 it is not clear if there is a gray scale, or all the non-zero elements are plotted as black points.
Page 9. Reference is made to “the unperturbed mode with m = 1 whose maximum is at the billiard center, y = Ly/2 “. There is no plot or reference supporting this statement.
Page 10: Fig. 6. The scale of the color code for the squared amplitude of the wave function (probably 10^(-3)) is missing.
Page 11, In Eq. (20) the comma is too close to P, and it looks like P´.
Page 13. “scare-like states” should be “scar-like states”
Author Response
We thank the Reviewer for recommending the publication of our paper, as well as
for reading it carefully. We also appreciate the suggestions, as they had helped
us to improve the presentation of our work.
Specifically, we made the following corrections to the manuscript in order to
account for all the suggestions (all changes are shown in red text for easy
identification):
1. In page 2 after “It can be seen that in this case the eigenfunctions are not
only ergodic but also maximally chaotic (random)” we cited new Ref. [10].
2. In page 2 after “the typical mechanism for scars is the presence of local
symmetries of the system, which can correspond to the presence of local integrals
of motion” we cited Refs. [20,21] where Ref. [21] is a new reference added to the
previous version of the paper.
3. In page 3 we rewrote some phrases such that the description is clearer.
4. We added information about the coefficients A_k in the caption of Fig. 1.
5. In the captions of Figs 2 and 3 we indicated that the matrices are shown in gray scale.
6. In page 9 after “the unperturbed mode with m = 1 whose maximum is at the billiard center, y = Ly/2" we wrote a sentence that supports the statement.
7. In the caption of Fig. 6 we indicated the correct color code scale.
8. In page 11 we deleted the coma in Eq. (20).
9. In page 13 we wrote “scar-like states”.
In addition we updated Ref. [19].
We hope that the revised version of our manuscript can now be published
in Entropy.
Reviewer 2 Report
In this paper, the authors study scaring effects in a rough rectangular billiard. Two types of scaring eigenstates were found numerically, one similar to the well-known coordinate-space ones, while, the other related to certain momentum space. One main observation is that the rough boundary may repulse some observed scaring eigenstates. To explain this fact, the authors reformulate the system by rewriting the one-particle rough-billiard Hamiltonian as that for two interacting particles in a flat billiard. This technique indeed brings insights into the problem, in particular, it enables one to study structure of the Hamiltonian in an unperturbed basis, which as is well known may supply useful information in properties of eigenstates. Besides, the introduction contains insightful comments concerning the recently-fast-developing field of scar and localization. Such analytical understanding of numerically obtained results in the field of scar is useful, because this is usually not achievable in this field.
To summarize, the main results are interesting and the paper is well written. I recommend its publication in Entropy.
Author Response
We thank the Reviewer for recommending the publication of our paper, as well as
for reading it carefully.